# Endoscopic Image Classification Based on Explainable Deep Learning

**DOI:** 10.3390/s23063176

**Published:** 2023-03-16

**Authors:** Doniyorjon Mukhtorov, Madinakhon Rakhmonova, Shakhnoza Muksimova, Young-Im Cho

**Affiliations:** Department of IT Convergence Engineering, Gachon University, Sujeong-Gu, Seongnam-Si 461-701, Republic of Korea; doniyorramsey@gachon.ac.kr (D.M.); madina182601@gachon.ac.kr (M.R.)

**Keywords:** explainable ai, deep learning, classification, endoscopic image

## Abstract

Deep learning has achieved remarkably positive results and impacts on medical diagnostics in recent years. Due to its use in several proposals, deep learning has reached sufficient accuracy to implement; however, the algorithms are black boxes that are hard to understand, and model decisions are often made without reason or explanation. To reduce this gap, explainable artificial intelligence (XAI) offers a huge opportunity to receive informed decision support from deep learning models and opens the black box of the method. We conducted an explainable deep learning method based on ResNet152 combined with Grad–CAM for endoscopy image classification. We used an open-source KVASIR dataset that consisted of a total of 8000 wireless capsule images. The heat map of the classification results and an efficient augmentation method achieved a high positive result with 98.28% training and 93.46% validation accuracy in terms of medical image classification.

## 1. Introduction

Artificial intelligence (AI)-based medical applications have replaced old traditional diagnostics with advanced detection and classification methods in recent years. AI has had a significant impact on medical imaging, as AI algorithms can assist radiologists in analyzing medical images, reduce the risk of misdiagnosis, and increase diagnostic accuracy. Also, AI algorithms can help automate routine tasks, such as image segmentation and analysis, freeing up the radiologists’ time for more complex tasks. In addition, AI algorithms can quickly analyze large amounts of medical image data, reducing the time required to obtain a diagnosis. As several proposals have used it, AI has reached sufficient accuracy to implement [1,2,3,4]; however, the algorithms are black boxes and are, therefore, difficult to understand without explanation. Deep learning algorithms, while highly effective in many applications, have several limitations when it comes to explainability. For example, deep learning algorithms can be difficult to understand and interpret, making it challenging to explain their decisions and understand how they arrived at a certain result; in addition, deep learning algorithms can inadvertently incorporate biases into the training data, leading to biased or discriminatory results. Also, the internal workings of deep learning algorithms are often considered a “black box”, making it difficult to understand why a certain decision was made. Without explainable deep learning, it can be challenging to understand why a deep learning model made a certain prediction or decision, making it difficult to identify potential problems and apply improvements.

Deep learning has achieved high performance for numerous types of endoscopic diseases when the number of images in the dataset is high enough during the training time. Computer vision is one of the main branches of deep learning, which is powered by classification, object detection, and microscopic image analysis in medical imaging. The main challenge of deep learning is overfitting when the dataset size is limited. To overcome this problem, collecting and augmenting the data are essential parts of deep learning before training and testing the model architecture [5]. In addition, there are a number of useful methods to overcome overfitting and increase the accuracy of the model, which are denoising [6], initialization and setting momentum [7], batch normalization [8,9], dropout [10], and drop connect [11]. Transfer learning is also a great technique to enhance the deep learning model with a high positive benefit in medical imaging using pre-trained weight in the last part of the convolutional neural network [12,13,14]. Traditional deep learning trains each model on a specific domain, whereas transfer learning uses natural image datasets or similar types of medical image datasets with pre-trained weight to reduce the training time. Feature extraction and fine-tuning are the most used types of methods for transfer learning in deep learning. Moreover, custom versions of the CNN are a common way to improve model architecture and increase the accuracy of the model by using different types of layers: dropout, dense layer, average pooling, fully connected layer, etc. [15,16,17]. AlexNet [18], ResNet [19], and VGG [20] are the original versions of the CNN, and they are modified with a number of changes in different fields for increasing to deep networks or reducing to lightweight networks.

Endoscopic image classification using explainable AI involves the use of machine learning algorithms to classify images taken during an endoscopic procedure and provide explanations for the classifications made by the model. This can be useful for identifying and diagnosing abnormalities or conditions within the body during the endoscopic procedure and for providing transparency and accountability in the decision-making process. The explainable AI approaches used include methods such as feature attribution, model interpretation, and rule-based systems [21,22]. Improving the confidence in AI model decisions will ensure a deeper understanding of model architecture and improve model performance. Explanation auxiliary handles clarify the decisions made by explainable models, making them more intuitive to humans. Further explanation of the decision made can increase the reliability of the method, thus helping healthcare professionals obtain the correct diagnosis. This requires explainable visualizations, explainable decisions, and mapping systems. XAI is a methodology and set of processes in which human users understand and trust the AI decision. Interpretability methods that open black-box models as an explainable algorithm are divided into local and global; local explains a single prediction, and global explains the overall model. Input types, such as tabular, text, image, and graph, are used with two types of XAI: model specific and model agnostic. Model specific can be used for one model or a group of models. Model agnostic can be applied to all models regardless of the model. Gradient-based methods are widely used to visualize neural networks, especially for explainable models, for example, Deep SHapley Additive exPlanation (SHAP); this model is a type of model-agnostic method that increases the interpretability of neural network results and transparency. One more example is the Local Interpretable Model-Agnostic (LIMA) that is used to process the future importance of the model to visualize the decision of the model.

Gastric cancer is one of the most common types of cancer in the world. As a result, the mortality rate is high, according to the systematic analysis of the global burden of disease study [23]. A large number of people die from a lack of early detection, and approximately 70% of patients miss optimal treatment because their lesion is too small to find in time. A number of approaches have reached high detection and classification accuracy by using a deep learning model, but we propose a new approach to classify an endoscopic image by using the Inception-Resnet-v2 [24], ResNet-50 [25], MobileNetV2 [26], ResNet-152 [27], and VGG16 [28] models with a Grad–CAM model specific for explainable artificial intelligence. In addition, we use a data augmentation method to increase the medical image efficiency and use a noise reduction method to overcome the overfitting problem while using a small dataset.

## 2. Background

The use of a dependable method to detect endoscopic disease is limited because of high-quality doctors and the time-consuming process to detect a single disease. The early detection of lesions and disease has several options, including object detection, classification, and segmentation. For example, the most used method is the CNN-based model of Kvasir [29], an open-source dataset to classify lesions, that has wireless capsule images trained with ResNet [30] and DenseNet [31]. The latest localization method is segmentation; the ground truth labeled image has to train as part of the supervised learning method. Thresholding [32] is one of the easiest versions of segmentation, where a threshold is set for dividing pixels into 0 and 1. Deep learning-based segmentation models are also common among endoscopic imaging, such as U-Net [33], SegNet [34], DeepLab [35], Mask-R-CNN [35], and other custom versions.

Hirasawa et al. [36] used 13,000 endoscopic images of the stomach, which included both normal and cancerous images, to train and evaluate the CNN model. The study found that the CNN model was able to accurately detect gastric cancer in endoscopic images with a high degree of sensitivity and specificity. Besides a 92.2 % sensitivity, the detection time was faster than 47 s. Cao et al. [37] presented a study in which the authors used a machine-learning model called Mask R-CNN to detect gastric cancer in endoscopic images. The study used a dataset of endoscopic images of the stomach, which included both normal and cancerous images, to train and evaluate the Mask R-CNN model. The authors found that their model was able to achieve high accuracy in detecting gastric cancer in endoscopic images. They also experimented with using a combination of the Mask R-CNN model with other image processing techniques to improve the performance of the model and found that this approach further increased the accuracy of the model. Li et al. [38] modified a combination of the CNN and magnifying endoscopy with narrow band imaging (M-NBI) and achieved 90.91% accuracy for diagnosing early gastric lesions. The dataset for the model was low-quality images, which was a disadvantage of the system. In addition to the above limitations, a small dataset may result in overfitting and a lack of accuracy in terms of the training process to overcome and enhance the model architecture. Shichijo et al. [39] introduced a pre-trained model consisting of a 22-layer CNN with a total of 32,208 images including fine-tuning. The images were classified into eight classes. As a result of transfer learning, the diagnostic time decreased considerably. Nakashima et al. [40] used GoogLeNet architecture as a pre-trained CNN model, created using a large number of public open-source images such as ImageNet. They increased the dataset capacity with data augmentation to identify positive and negative cancer. The experimental result showed that the AUC, sensitivity, and specificity of the model were 66.0%, 96.0%, and 95.0%, respectively. A small polyp is hard to detect in the early stage, but it can cause serious and dangerous illnesses. Tajbakhsh et al. [41] constructed a combination of CNNs that were used to classify the interesting regions to obtain image features including color, texture, shape, and temporal information. The initial information and image trained with the combined CNN as a three-dimensional neural network reduced the number of false predictions even for a small polyp.

A broad approach to deep learning has yielded very positive results both in terms of classification and detection of endoscopic images, but in most studies using black-box deep learning systems, AI decisions are hidden. Gradient explanation techniques [42,43] are gradient-based attribution methods that explain the decision of deep learning using heat mapping. Grad–CAM [44] is a new method of combining feature mapping using gradient signals that use gradient information flowing into the CNN final convolutional layer to assign important values to each neuron for the determination of particular interest. Grad–CAM is called the last convolutional layer here, but it is a CAM that can be applied to any layer to create a visual description of any CNN regardless of its architecture, which is one of the limitations of CAM. A limitation of Grad–CAM is its inability to localize multiple occurrences of an object in an image due to partial derivation assumptions. Hires–CAM [45] is an improved version of Grad–CAM with modifications of the limitations in the averaging step. The proposed high-resolution class activation mapping method does not average over gradients; instead, multiple feature mapping is directly calculated. Grad–CAM++ is similar to Grad–CAM, but visualization of the object classification is strong because of a more sophisticated algorithm with increasing weight size. Score–Cam [46] is one of the smoother versions of Grad–Cam that has more consistent and less noisy heat maps. Score–Cam is obtained with a linear combination in the last stage for giving a visual explanation. Layer–CAM [47] uses pixel-wise weights to generate and integrate convolutional neural networks, and it indicates a key part of object heat map classification. Xgrad–CAM [48] typically propagates the class score at the classification layer to the convolutional layer and generates object localization in weakly labeled remote sensing. Grad–CAM and Grad–CAM++ are methods for visualizing the regions of an image that are most important for a given classification decision made by a convolutional neural network (CNN). They are commonly used to provide explanations for the decisions made by a CNN and can be used to analyze endoscopic images for the task of medical image classification. Grad–CAM uses the gradients of the output of a CNN with respect to the input image to generate a heat map that highlights the regions of the image that contributed the most to a given classification decision. Grad–CAM++ is an extension of Grad–CAM that improves its ability to highlight the relevant regions of the image by taking into account the gradients of the output of the CNN with respect to both the input image and the intermediate activations of the CNN. Both methods have been applied to endoscopic image classification tasks, such as detecting polyps and tumors in colon images and gastric cancer in stomach images. These visualization techniques can help in understanding the features that the CNN model is looking at when making a classification decision and can provide insight into which regions of the image are most important for detecting a specific condition.

## 3. Method and Materials

### 3.1. Study Design

This study was conducted to classify wireless endoscopic images using explainable deep learning models. The aim of our study was to develop both model accuracy and efficient data augmentation and create explainable architecture. Supervised learning was selected as the DL method. We compared a number of DL models to get the best model among several models, such as lightweight, transfer learning, etc. The structure of the main method (Figure 1) is summarized as follows:Data augmentation method is used to increase dataset size and efficiencyTrain and test the dataset into several chosen CNNs to compare the best option for the classification of wireless endoscopic images to choose as a pre-trained modelThe third and last stage is to contribute to the classification of localized explainable areas using gradient explanation

### 3.2. Dataset

The Kvasir [48] dataset is a publicly available dataset for endoscopic image classification that contains images of the digestive tract taken during upper and lower gastrointestinal endoscopies. The dataset includes over 8,000 images of the esophagus, stomach, and colon along with annotations indicating the presence of various conditions, such as polyps, ulcers, and inflammation. The dataset includes different resolution images from 720 × 579 to 1920 × 1070 pixels and is divided into 8 classes, including z-line, cecum, esophagitis, polyps, ulcerative colitis, dyed resection margins, and dyed-lifted polyp. Figure 2 shows sample images of the dataset with class names. The dataset is split into train, validation, and test as an 8:1:1 ratio from the full dataset with 6400 images for train, 800 images for validation, and 800 images for test.

### 3.3. Data Augmentation

CNN-based classification requires thousands of measurements to train the data. Data augmentation is an essential method to solve the problem of processing a limited number of medical images and overcome the overfitting problem. Traditional data enhancement, such as color change images, random rotation, and rotation, can be greatly changed for medical images because the sensitivity of medical images is higher than general images. We decided not to use the colors directly because overfitting occurs during training if the color of the image changes rapidly. For the above reasons, we tried to use different approaches to solve the problem with brightness randomly selected for colors > 2 (Figure 3). In Equation (1), hue > 2 is a parameter that affects brightness and contrast. *F*(*x*) is the source pixel intensity, and *g*(*x*) is the output pixel intensity:(1)$$g\left(x\right)=\alpha (f\left(x\right)+\beta$$

The above Equation (1) represents the equation of rotation for our augmentation. The new coordinates *X* and *I* are obtained from the random angle formula.
(2)$$\left(X\right)=\cos\left(angle\right)X-\sin\left(angle\right)I$$
(3)$$\left(I\right)=\sin\left(angle\right)X+\cos\left(angle\right)I$$

### 3.4. Classification of the Type of Gastrointestinal Disease Based on the CNN

In the initial stage, we used a customized CNN architecture to classify the images for gastrointestinal disease. We customized several existing CNN models to obtain a CNN model for our architecture. There are several successfully used CNN models that have been developed for the classification of wireless endoscopic images, such as ResNet-18, ResNet-152, MobileNetv2, DenseNet201, and VGG16, that we implemented with new augmentation and configurations. Two types of the most commonly used ResNet models were chosen for training and comparison with other architectures: VGG16, MobileNetv2, and DenseNet201. Among these models, ResNet-152 showed more accurate results when training and testing our dataset. We fine-tuned the network’s hyper-parameters and set them as Table 1. First, we chose the number of epochs based on early stopping techniques; after 100 epochs, the accuracy remained stable. The bath size was selected as 64 because of our GPU capability, as we used a 24 GB graphics card. Choosing an appropriate learning rate is a crucial step in training a convolutional neural network (CNN). The learning rate determines how much the weights of the network are updated in response to the loss gradient during backpropagation. If the learning rate is too high, the training may diverge or fluctuate wildly. If the learning rate is too low, the training may take a long time to converge or get stuck in a local minimum. We adjusted the learning rate based on the training progress and the observed validation loss and adjusted the learning rate decay in the same way. The number of epochs before decay determines how long the network will train with a constant learning rate before decreasing it. We started with a relatively large number of epochs before decay, such as 50, but we decreased by 10 because of our dataset capacity.

ResNet and VGG are similar CNN networks, but VGG is deeper, has more parameters, and uses only 3 × 3 filters. ResNet-18 and ResNet-152 are both deep convolutional neural network architectures that use residual connections to allow for very deep networks to be trained. However, there are significant differences between the two in terms of their depth, number of layers, and overall complexity. ResNet-18 has a relatively small number of parameters (11 million) and is commonly used for smaller-scale image classification tasks. ResNet-152 is a much deeper network than ResNet-18 with a total of 152 layers and also consists of residual blocks but with significantly more layers per block (up to 50) and more filters per layer; it also has a bottleneck architecture that reduces the dimensionality of the feature maps in the middle of the block. ResNet-152 has a much larger number of parameters (60 million) and is typically used for more complex tasks. DenseNet201 is a deep convolutional neural network architecture similar to ResNet and other deep networks but with a unique dense connectivity pattern that allows for a more efficient use of parameters and better performance on image classification tasks. DenseNet201 consists of 201 layers and has approximately 20.6 million parameters. The network is composed of several dense blocks, where each block contains multiple layers that are densely connected to each other. Specifically, in each dense block, each layer receives feature maps from all preceding layers in the block as inputs, and its own feature maps are passed on to all subsequent layers in the block. This creates a dense connectivity pattern that enables the network to extract more useful features from the input image.

### 3.5. Visualization of Classification with Explainable AI

In this section, explainable AI techniques are used for the classification of wireless endoscopic images. We implemented a number of XAI techniques to get the optimal model and high results to compare with each model. The following methods are Grad–CAM, Hires–CAM, Layer–CAM, Grad–CAM++, Xgrad–CAM, and Score–CAM.

Gradient-weighted class activation maps (Grad–CAM) are one of the most used XAI techniques. The technique gets the output as input before the classification block in the CNNs. The general concept of Grad–CAM includes calculating the gradient of a given output neuron and computing this gradient backward towards the last convolutions. The layer before classification saves the most detailed information learned with the CNN. Three methods are included to obtain the model heat map described as follows: gradient calculation, alphas calculation via averaging gradients, and the final Grad–CAM heat map calculation. First, compute the gradient of $y_{c}$ with feature map activation $A^{k}$ of the convolutional layers. The value of a particular gradient in Equation (4) is equal to the input image because the image input represents the feature map as well.
(4)$$ComputedGradient=\frac{\delta y^{c}}{\delta A^{k}}$$

The second step is to calculate the alpha value by averaging a set of global variables against the value of the breadth measure *I* and the height dimension index *j* to obtain neuron importance weights $\alpha_{k}^{c}$ in Equation (5).
(5)$$\alpha_{k}^{c}=\frac{1}{Z}\sum_{i}\sum_{j}\frac{\delta y^{c}}{\delta A^{k}}$$

The last step performs a weighted combination of feature map activation $A^{k}$, where the weights are the $\alpha_{k}^{c}$ added to calculate the final value. The alpha value calculates a weighted sum of feature maps as the last Grad–CAM heat map; then, a ReLU operation uses only the positive values and turns all the negative values into 0 as Equation (6).
(6)$$L_{Grad-CAM}^{c}=ReLU(\sum_{k}\alpha_{k}^{c}A^{k}$$

Although Grad–CAM is the most used version of XAI methods for medical imaging so far, Grad–CAM heat maps are not able to reflect the computation of the model, which may cause irrelevant areas while predicting specific areas. The Hires–CAM avoids this problem with an element-wise product between the raw gradients and the feature maps. The initial step is the same as with Grad–CAM, calculating $\frac{\delta y^{c}}{\delta A^{k}}$, in the further step, and the interest map is obtained by elementally multiplying the gradient and feature maps before they are combined for the feature measure.

## 4. Experimental Results

The experimental results were conducted on our laboratory’s PC with an AMD Ryzen 12-core processor of 3.70 GHz, 64 GB RAM, and a 24 GB graphics card. First, we compared several CNN models to get the prior backbone method for the explainable AI method. We set pre-processing steps, such as data augmentation, train configurations, etc. There are several ways to compare the models, such as performance metrics; one of the most common ways to compare machine learning models is to evaluate their performance using relevant metrics, such as accuracy, precision, recall, and F1-score. These metrics provide a quantitative assessment of the model’s ability to solve a specific problem. Accuracy is the proportion of correct predictions made by the model. Precision (7) is the proportion of true positive predictions made by the model among all positive predictions. Recall (8) (Sensitivity) is the proportion of true positive predictions made by the model among all actual positive cases. Eventually, F1-score (9) is the harmonic mean of precision and recall. Overall, it is important to consider a combination of these factors when comparing machine learning models to determine which model is best suited for a specific problem.
(7)$$\mathrm{Precision}=\frac{\mathrm{TP}}{\mathrm{TP}+\mathrm{FP}}$$
(8)$$Recall=\frac{TP}{TP+FN}$$
(9)$$F1=2\times \frac{Precision\times Recall}{Precison+Recall}$$

Initial results showed that ResNet-152 obtained the highest train results in terms of train accuracy with 98.28% (see Figure 4 and Figure 5).

Among all of the tested CNN methods, we used 100 epochs per model and calculated the training time. The lightweight MobileNetv2 obtained the lowest training time with only 6.1 h; however, DenseNet201 finished the 100 epochs in 11.9 h. Figure 6 shows detailed information on the experimental results of the CNN training with the confusion matrix.

As the confusion matrix of the CNN models represents, ResNet-152 obtained the highest accuracy in terms of the classification of the endoscopic images and took 10 h for training with the Kvasir dataset. Mobilenetv2 obtained the trained weight in 6.1 h. Table 2 shows detailed information of the experimental results of the CNN.

As we mentioned above, medical data augmentation was implemented to avoid overfitting problems and increase data set capacity. Figure 7 shows the difference between our data augmentation and without changing the original dataset.

ResNet-152 was chosen to compare data augmentation, and the model achieved the highest score in terms of both training and validation. Table 3 indicates that our new data augmentation reached better performance in both training and validation with 98.28% and 93.46%, which was 6.28% and 3.7% higher than without using data augmentation.

The last part of the experimental result is comparing the output of the gradient heat map from our chosen explainable methods. The Hires–CAM and Grad–Cam showed better results than the other methods to visualize the heat map. Figure 8 shows the performances of the explainable AI.

The output of the classification heap map shows that all types of explainable models, including the Grad–CAM, Grad–Cam++, Layer–Cam, Hires–Cam, and Xgrad–CAM, have almost similar results, whereas Grad–CAM++ obtained higher heat maps on both dyed-p and polyp classes. As dyed-p consists of more fixtures in the image, it has a more accurate heat map than the polyp class.

Table 4 shows the classification results of other related research with the Kvasir datasets, whereas we implemented not only a classification task but also explainable heat mapping results in this study. Among the research, our proposed method obtained higher classification accuracy.

## 5. Conclusions

Explainable deep learning-based classification is essential for endoscopic imaging because it can help in interpreting the decisions made by the deep learning model. In medical applications, it is crucial to understand how the model makes decisions so that healthcare providers can explain the results to their patients and make informed decisions about the diagnosis and treatment. Explainable deep learning-based classification models can provide insight into the features of the endoscopic images that contribute to the model’s decisions, making it easier to understand the reasoning behind the model’s predictions. Furthermore, explainable models can help detect potential biases or errors in the model’s decision-making process. By understanding how the model works, healthcare providers can identify potential issues and take steps to mitigate them.

This study proposed an explainable artificial intelligence method for wireless endoscopic image classification using ResNet-152 combined with Grad–CAM. We used an open-source KVASIR dataset for both data augmentation and a custom CNN model. The model was improved thanks to the explainable artificial intelligence (XAI) perspective that was conducted via numerically updated data with a new data augmentation method and using a heat map explanation of the classification, such as Grad–CAM, Grad–Cam++, Layer–Cam, and other methods. In addition, our method conducted the custom CNN training and testing parts for comparing a number of CNN methods with custom configurations. Explainable deep learning, also known as interpretable machine learning, is particularly important in the context of wireless endoscopic images because it allows medical experts to understand and validate the decisions made by the model. This is particularly important in the field of medical imaging, where the decisions made by the model can have a significant impact on a patient’s health. One key advantage of using explainable deep learning in the context of wireless endoscopic images is that it allows medical experts to identify and diagnose abnormalities and conditions within the body during the endoscopic procedure. This can improve the diagnostic accuracy of endoscopic examinations and lead to earlier detection and treatment of diseases.

The experimental results showed that the proposed method was implemented as we expected; we observed an improved model performance, with 98.28% training and 93.46% validation accuracy, and compared five explainable methods as well. The research helps to open the black box of deep learning performance in endoscopic images. The advantages of our conducted method are the comparison of several CNN models for getting the best option, and the model was used with a combination of Grad–CAM and an efficient medical data augmentation method to improve the accuracy of classification from 89.29% to 93.46% accuracy. In addition, we tried five different heat map methods, Grad–CAM, Grad–CAM++, Layer–CAM, Hires–CAM, and Xgrad–CAM, to compare the output decision of the classification. The research shows the advanced fine-tuned parameters for the Kvasir dataset that we chose for the CNN models. When it comes to limitations, we brought only a new data augmentation method and used existing methods to construct a combined method based on ResNet-152 and Grad–CAM instead of constructing a totally new architecture.

According to the obtained findings and results, the developed solution provided positive performance outcomes regarding endoscopic image classification in the above study. There are limitations, including a lack of updated open-source datasets in wireless endoscopic images. The authors already planned the future direction to improve model performance to modify new updates in further works.

Overall, the use of explainable deep learning in the context of wireless endoscopic images can improve the diagnostic accuracy of endoscopic examinations, provide transparency and accountability in the decision-making process, and aid the medical expert to understand the decision made by the model, even if they are not able to inspect the images themselves.

## Figures and Tables

**Figure 1 sensors-23-03176-f001:**
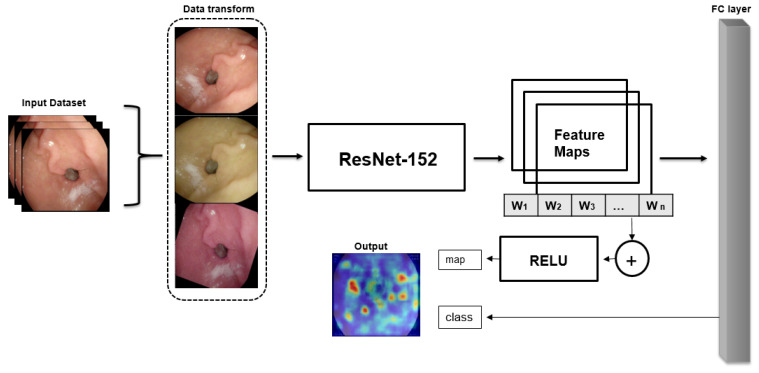
Overall architecture of the explainable AI for the classification of endoscopic images.

**Figure 2 sensors-23-03176-f002:**
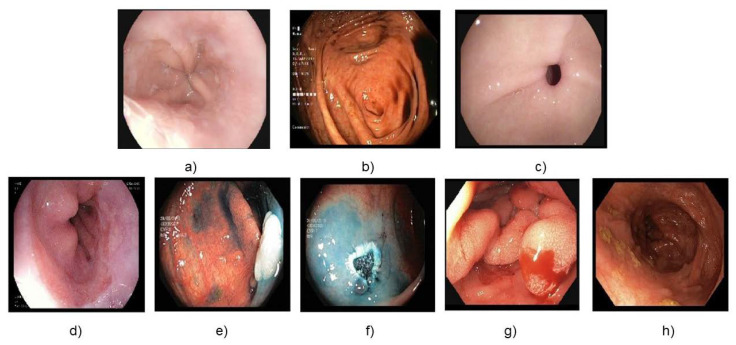
Dataset classes: (**a**)normal z-line, (**b**) normal cecum, (**c**) normal polorus, (**d**) esophagitis, (**e**) dyed and lifted polyps, (**f**) dyed dissection margins, (**g**) polyps, and (**h**) ulcerative colitis.

**Figure 3 sensors-23-03176-f003:**
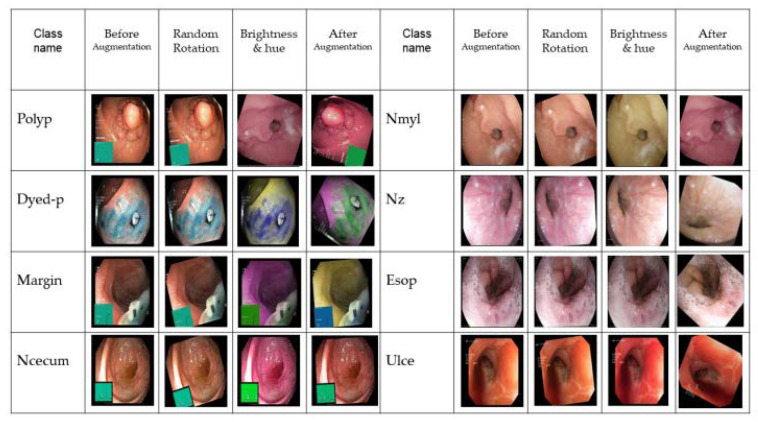
Data augmentation methods and the result of augmentation.

**Figure 4 sensors-23-03176-f004:**
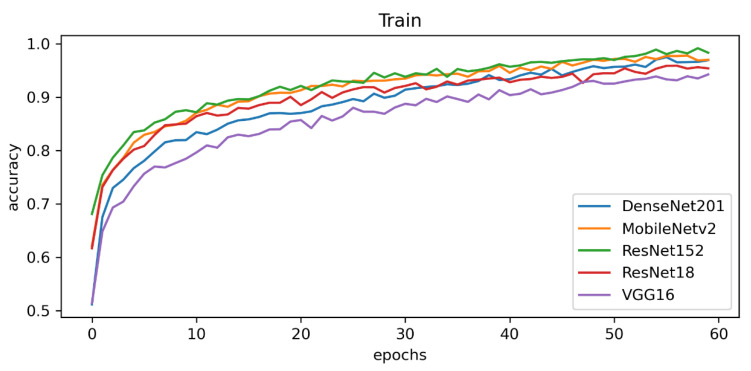
Train accuracy of the CNN models for the backbone of explainable AI.

**Figure 5 sensors-23-03176-f005:**
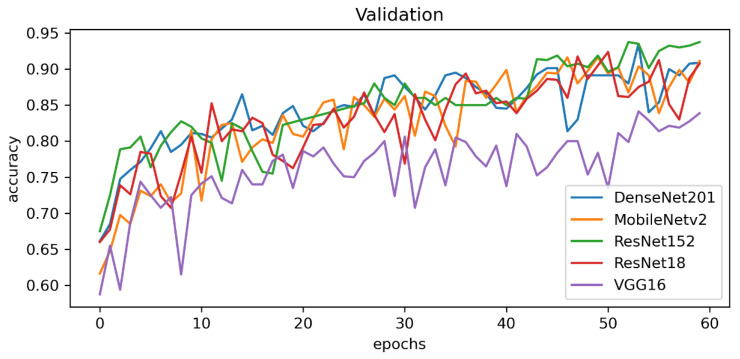
Validation accuracy of the CNN models for the backbone of explainable AI.

**Figure 6 sensors-23-03176-f006:**
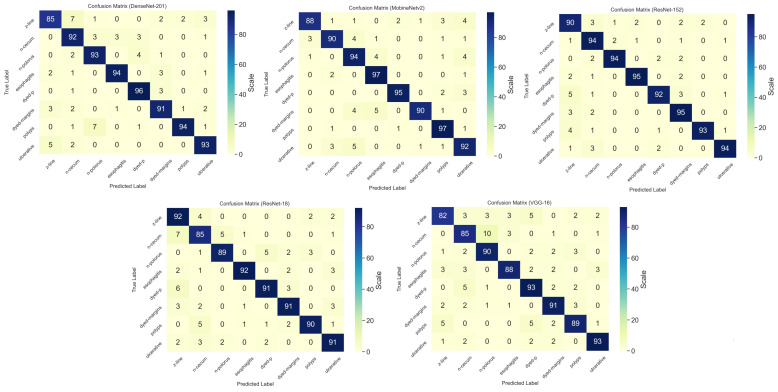
Confusion matrix of the CNN models with true and predicted class.

**Figure 7 sensors-23-03176-f007:**
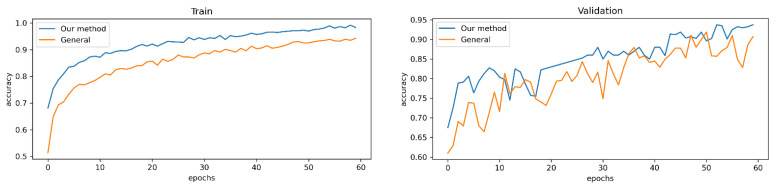
Data augmentation training and validation results.

**Figure 8 sensors-23-03176-f008:**
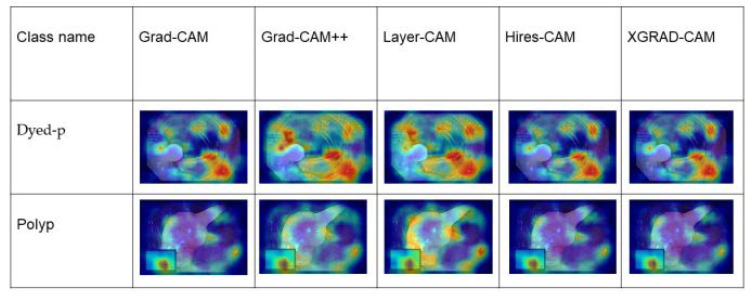
The output of the explainable heat map for dyed-p and polyp.

**Table 1 sensors-23-03176-t001:** Custom hyper-parameters for the CNN model.

Type of Hyper-Parameter	Value
number of epochs	100
batch size	64
learning rate	0.001
learning rate decay	0.9
number of epochs before decay	10

**Table 2 sensors-23-03176-t002:** Detailed information on the training results of the CNN models.

Method	Train Accuracy (%)	Train Loss	Validation Accuracy (%)	Validation Loss	Epoch	Train Time (h)
DenseNet201	97.92	0.07	92.29	0.23	100	11.9
MobileNetv2	96.46	0.1028	92.37	0.18	100	6.1
ResNet-152	98.28	0.055	93.46	0.11	100	10
ResNet-18	96.03	0.1129	90.12	0.23	100	5.6
VGG16	96.15	0.101	88.26	0.26	100	8.25

**Table 3 sensors-23-03176-t003:** Data augmentation method performance.

Method	Train Accuracy (%)	Train Loss	Validation Accuracy (%)	Validation Loss	Epoch
General	91.92	0.27	89.29	0.23	100
Our method	98.28	0.055	93.46	0.11	100

**Table 4 sensors-23-03176-t004:** The results of comparison experiments.

Author	Model	Dataset	Accuracy
Sandler et al. [49]	MobileNetv2	Kvasir	79.15 %
Borgli et al. [50]	Teacher–student Framework	HuperKvasir [51]	89.3 %
Srivastava et al. [52]	FocalConvNet	Kvasir	63.7
Pozdeev et al. [53]	Custom CNN for two-stage classification	Kvasir	88.00%
Fonolla et al. [54]	Multi-model Classification	Kvasir	90.20%
Our method	ResNet-152 combined with Grad–CAM	Kvasir	93.46 %

## Data Availability

Open-source Kvasir dataset is available on https://datasets.simula.no/kvasir/ (accessed on 10 March 2023).

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
