# Peer review of "Endoscopic Image Classification Based on Explainable Deep Learning"

_sensors, 2023, doi:10.3390/s23063176_

Round 1
Reviewer 1 Report
1. The introduction of the model is not clear. There is no detailed introduction of the algorithm model used in the article. There is no introduction to the designed model framework.
2. There is no corresponding experiment for the design of the weight parameters of the third and last stages. There is no good explanation for setting those weights.
3. There are no comparison indicators for classification results. At the same time, accuracy values for each category are missing.
Author Response
Dear reviewer, after getting your comments, we modified our manuscript by adding more detailed information. Thank you for your contribution.

Reviewer 2 Report
-The paper should be interesting ;;;
-it is a good idea to add a block diagram of the proposed research/review (step by step);;;
-it is a good idea to add more photos of measurements, sensors + arrows/labels what is what (if any);;;
-What is the result of the analysis?;;
-figures should have high quality. ;;;;;
-text should be formatted;;;;
-please add photos of the application of the proposed research, 2-3 photos ;;;
-what will society have from the paper?;;
-labels of figures should be bigger;;;;
-please compare the advantages/disadvantages of other approaches;;;
-Conclusion: point out what have you done;;;;
-please add some sentences about future work;;;
-references to Tables and Figures should be added in the text for example ref. to Fig. 3 is missing;;
Author Response

(The authors gave the same response as above.)

Reviewer 3 Report
The article proposes Endoscopic Image Classification based on Explainable Deep Learning. The article cannot be accepted in its current form. However, I propose to improve the article by addressing the following points:
1. Numerical/Statistical results are to be added to the abstract.
2. The abstract needs to be improved by highlighting the novelty of the work.
3. Line 208-210 “CNN-based classification requires thousands of measurements to train the data. Data augmentation is an essential method to solve the problem of processing a limited number of medical images to overcome the overfitting problem.”
I do not see a valid reason to support the statement above. From line 202, it is found that the author utilized 6400 images for training and it is more than enough to train any CNN model. What is the need for data augmentation?
4. Line 227-228 "... We customized several existing CNN models to obtain a unique CNN model for our architecture."
I found no technical details about the unique CNN model employed in this paper, but the authors just mention customized versions of several existing CNN models.
5. Line 234-235 "... We fine-tuned the network’s hyperparameters and set them as Table 1.."
There is just a brief mention of the fine-tuned network's hyperparameters, but there is no detail on the fine-tuning process used and how the value shown in table 1 was fixed.
6. From figures 4 and 5, it appears that the author used 100 epochs. But as per table 1, the number of epochs is fixed at 200. There is a mismatch between the parameters set for the network and the parameters used for analysis.
7. From Figures 4, 5 and 6 I do not find any change in accuracy after 60 epochs. What is the need for using 100 epochs?
8. Instead of using more than 60 epochs the author should change the learning rate to obtain a better result.
9. There are no proper explanations provided about the number of layers used in the CNN Learning Models. Details like the number of layers, the number of neurons per layer, and the number of training iterations are to be added to the manuscript.
10. Authors need to bring novelty and originality to their review work. They need to establish the clear superiority of their proposed methodology through comprehensive comparison results with very recent algorithms
11. The author has presented well-recognized and published methodologies with some applications. A constructive conclusion regarding the proposed model is not found in this article.
12. The contribution of the conclusion section is very limited. In the proposed model, there is a lack of explanation in detail as to why better results are obtained.
13. The comparison work is extremely weak. More technical comparisons with other existing methods should be provided.
Author Response

(The authors gave the same response as above.)

Round 2
Reviewer 1 Report
1. Is Formula 7 the same as Formula 8? Is it wrong?
2. The quality of the picture is very poor. In particular, the picture quality of Figure 6 is very poor, and the horizontal and vertical coordinates are very fuzzy. Figure 8 is also of poor quality.
3. The header of Table 2 is separated with the table , so the format of the paper must be revised.
4. Table 4 lists the data set KID, but it is not clear what is the difference between the data set KID and the data set Kvasir, and why those two data sets should be compared in one table.
5. Which of formulas 7, 8 and 9 does the accuracy in Table 4 refer to? The paper is not rigorous in many aspects.
6. In the experiment, all methods are compared with the mean value of precision. Why not list the precision of each type of Gastrointestinal disease in a table, and then analyze the effectiveness of the method?
7. There is still no detailed description of the model architecture, and the innovation of the method is not reflected.
Author Response
Dear Reviewer,
We are writing in response to your review of our paper which was submitted to Sensors and published by MDPI. We would like to thank you for taking the time to review our paper and for providing us with valuable feedback.
We have addressed the editorial mistakes that were highlighted in your review and made necessary corrections to ensure that the paper adheres to the standards of academic writing. Additionally, we have trained our model with new epochs and updated our experimental results accordingly. We believe that these changes have significantly improved the quality of the paper.

Reviewer 2 Report
References, text, figures, and tables should be formatted according to MDPI format.
Formulas (1)-(9) should be formatted also.
Line 339 why so badly formatted?
"main method(Figure1) is summarized as follows" should be
"main method(Figure 1) is summarized as follows"
Author Response

(The authors gave the same response as above.)

Reviewer 3 Report
The author did not include an explanation for the following point
8. Instead of using more than 60 epochs the author should change the learning rate to obtain a better result.
I request the author try different learning rates and present the result.
Author Response

(The authors gave the same response as above.)
